# Water Conservation Methods and Cropping Systems for Increased Productivity and Economic Resilience in Burkina Faso

**Hamidou Traoré [1,\*], Albert Barro [1], Djibril Yonli [1], Zachary Stewart [2,3] and Vara Prasad [2,3]**

1  Institut de l'Environnement et de Recherches Agricoles, 04 BP 8645 Ouagadougou, Burkina Faso; altbarro@yahoo.fr (A.B.); d.yonli313@gmail.com (D.Y.)

2  Feed the Future Innovation Lab for Collaborative Research on Sustainable Intensification, Kansas State University, Manhattan, KS 66506, USA; zachstewart@ksu.edu (Z.S.); vara@ksu.edu (V.P.)

3  Department of Agronomy, Kanas State University, Manhattan, KS 66506, USA

\*  Correspondence: hamitraore8@yahoo.com; Tel.: +226-70258060; Fax: +226-25340271

**Abstract:** Resilience of smallholder farmers in their ability to bounce-back and overcome shocks, such as drought, is critical to ensure a pathway out of hunger and poverty. Efficient water conservation methods that increase rainwater capture and reduce soil erosion such as stone lines and grass bands are two technologies that have been proposed to increase the resilience in Sudano–Sahelian farming systems. In Burkina Faso, we show that stone lines, grass bands, and crop rotation are effective resilience strategies individually and in combination. During years when rainfall is well-distributed over time, differences are minimal between fields with water conservation methods and fields without. However, when there are periods of prolonged drought, water conservation methods are effective for increasing soil water, yield, revenue, and resilience. During drought conditions, sorghum (*Sorghum bicolor* (L.) Moench) grain yield and revenue with stone lines and grass bands were over 50% greater than that of the control, by an average of 450 kg ha$^{-1}$, which amounted to an increase of 58,500 West African CFA franc (CFA) ha$^{-1}$ (i.e., 98 USD ha$^{-1}$). The results also suggest that the combination of water conservation method and crop rotation additionally improves cropping system productivity and revenue. Growing cowpea (*Vigna unguiculata* (L.) Walp.) in rotation with sorghum production provided more options for farmers to increase their income and access to nutrition. This study also sheds light on the limited productivity gains due to improved crop varieties. The local sorghum landrace, Nongomsoba, and the local cowpea variety in rotation resulted in the highest yields as compared to the improved varieties of Sariaso 14 sorghum and KVX 396-4-4 cowpea. Under similar low input/degraded conditions, improved crop varieties likely are not a suitable resilience strategy alone. We conclude that during erratic rainy seasons with frequent periods of drought (i.e., water stress) in rain-fed conditions in Burkina Faso, stone lines or grass bands in combination with sorghum and cowpea rotation are effective practices for increasing resilience of smallholder farmers to maintain crop productivity and revenue. With future and present increases in climate variability due to climate change, stone lines, grass bands, and crop rotation will have growing importance as resilience strategies to buffer crop productivity and revenue during periods of drought.

**Keywords:** Stone lines; grass bands; resilience; cropping systems; crop rotation; food security; Sudano–Sahel

## 1. Introduction

Soil degradation is a major constraint in sub-Saharan Africa (SSA) [1]. Soil fertility is declining largely due to crop nutrient mining using traditional farming practices and soil erosion [2]. Recent

efforts to prioritize methods to improve soil fertility in West Africa have focused on system approaches that integrate inorganic and organic fertilizer approaches as well as socioeconomic considerations [3,4]. The removal of crop residues coupled with low soil-water holding capacity and low rates of fertilizer application significantly reduces the soil nutrient balance. In West Africa, 4.4 million tons of N are lost per year, and only 0.8 million tons of N are returned back to the soil [5]. Especially across the Sudan to Sahel regions, rainfall is low and erratic; however, in general, the water shortage is not driven by total water supply, but rather by the ability of the cropping system to retain water and reduce runoff following extreme and erratic rainfall events. With an average annual rainfall of around 800 mm, this is adequate to support the genetic potential of sorghum, millet, and cowpea production [6,7]. However, over the last 35 years, "extreme" [8] daily rainfall totals have tripled [9], giving increased urgency to integrating resilience strategies into farming systems. Under these conditions, most rainfall is lost through runoff, carrying with it arable soil. Soils in the Sudan to Sahel regions have limited soil-water holding capacity due to high levels of sand, hard surfaces due to laterite, or limited soil organic carbon that is usually under 0.05% [10,11]. These soil nutrient and water constraints contribute to low crop productivity and limited crop response to inorganic fertilizer. Cereal crops, particularly sorghum (*Sorghum bicolor* (L.) Moench) and pearl millet (*Pennisetum glaucum* (L.) R. Br.), and legumes such as cowpea (*Vigna unguiculata* (L.) Walp) are staple foods in SSA, especially in Burkina Faso [12]. Low and erratic rainfall, poor soil fertility, and lack of high-yielding and stress-tolerant varieties paired with lack of good agronomic practices are the primary causes for low productivity in Burkina Faso.

Research and indigenous knowledge has identified many water conservation and nutrient management technologies for Sudano–Sahelian farming systems [13–15]; however, their effect on cropping systems resilience is less known. Additionally, several improved crop varieties have been released for Sudano–Sahelian farming systems to increase productivity [16]. Soil and water conservation practices are critical to the resilience of these farming systems [3,4,17,18]. Climate change leading to extreme weather events (i.e., both too much and too little rainfall) often reduces the resilience of farmers whose livelihoods depend on these soils. Water management techniques, improved cropping systems, and improved crop varieties may improve resilience by reducing the impact of extreme weather events and thus improve the ability of farmers to bounce-back from shock (i.e., resilience). It was our hypothesis that practices that retain soil water during times of drought and reduce soil erosion/degradation during times of heavy rainfall, increase systems diversity using crop rotation, and utilize improved crop varieties will increase farmer economic and productivity resilience. Thus, the objective of this study was to evaluate the effect of stone lines and grass bands, crop rotation, and improved varieties of sorghum and cowpea, either integrated or separate, on soil water, grain and biomass production, and revenue. Additionally, our objective was to observe these parameters during periods of extreme weather events to evaluate their impact on the system's ability to bounce-back (i.e., resilience) following such a shock.

## 2. Material and Methods

### 2.1. Site Description and Experimental Design

Experimentation was conducted at the Saria Agricultural Research Station (12°16′ N, 2°9′ W, 300 m altitude) in Burkina Faso. The climate is north Sudanian [19], with an average annual rainfall of 800 mm (30 year average). Rainfall is monomodal, lasting for 6 months (May to October) and is distributed irregularly in time and space (Figure 1). Mean daily temperatures vary between 30 °C during the rainy season and 35 °C in April and May. The mean potential evapotranspiration is 2096 mm in dry years and 1713 mm in wet years [20]. The site was previously under fallow for 10 years and was an open woody savanna [19]. The soil type is Ferric Lixisol [21] with an average slope of 1.5% and a hardpan at a depth of 80 cm that limits root growth [22] The textural class according to the USDA system is sandy loam in the 0 ± 30 cm layer (62% sand, 28% loam, 10% clay) with a gravel content decreasing from 36% at the 0–5 cm layer to 30% at the 5–10 cm layer. Average bulk density is 1.7 g/cm$^3$ within the

0–15 cm depth. Soil in the 0–30 cm depth had 6 g kg⁻¹ of organic C, 0.5 g kg⁻¹ of N, 46 mg kg⁻¹ of exchangeable K, and 15 mg kg⁻¹ of available P. The pH decreases from 5.3 in the 0–20 cm soil layer to 4.9 in the 60–80 cm layer.

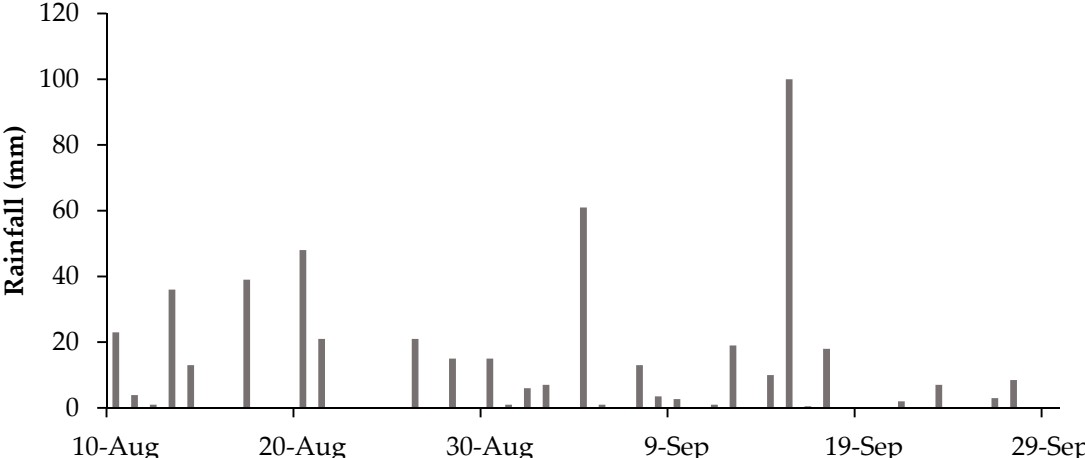

**Figure 1.** Rainfall distribution and quantity during the 2010 cropping season at Saria Research Station, Burkina Faso.

An integrated multifactor experiment was conducted over three seasons (2008–2010). The experimental design was a split-plot with water conservation method (WCM) treatments as the whole-plot and cropping systems and sorghum genotypes as subplots. Twelve treatments derived from the combination of the three factors were replicated four times. The factors and subfactors were as follows—water conservation method (WCM): 1. no WCM, 2. stone lines, and 3. grass bands of *Andropogon gayanus* (Kunth); cropping systems (CS): 1. continued sorghum, and 2. sorghum in rotation with cowpea; and genotype (G): 1. local sorghum landrace Nongomsoba or local cowpea variety, and 2. improved sorghum variety Sariaso 14 or improved cowpea variety KVX 396-4-4 (Figures 2 and 3). All plots were cropped in sorghum in 2008 and 2010. In 2009, for the crop rotation plots, a local cowpea variety was grown after sorghum landrace Nongomsoba while the improved cowpea variety KVX 396-4-4 was used after improved sorghum variety Sariaso 14. The improved varieties were reported to have enhanced yield and pest resistance. In each replication, WCM plots were separated by 1 m and the cropping system plots within the same WCM plot were separated by 1 m. Treatments were 40 × 6 m = 240 m² each. Sorghum and cowpea were sown 0.8 m between rows and 0.4 m between hills (plants). Two sorghum varieties (Sariaso 14 and Nongomsoba) were compared in continued cropping and rotation with a local or an improved cowpea variety (KVX 396-4-4). The cycle of Sariaso 14 varied between 115 and 120 days while that of Nongomsoba varied from 120 and 125 days. The cycle of the local and improved cowpea variety (KVX 396-4-4) is approximately 70 days. In all plots, planting for sorghum and cowpea was done in early July. Sorghum plots were fertilized at a rate of 100 kg ha⁻¹ with 14N-23P-14K-1B-6S and 50 kg ha⁻¹ of nitrogen was added as urea (46N). For cowpea, only NPK (14-23-14) was applied at a rate of 100 kg ha⁻¹.

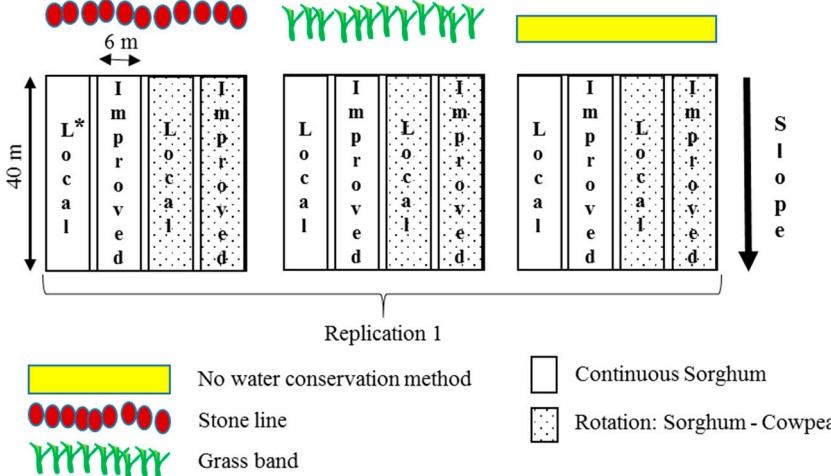

**Figure 2.** Experimental design. There were four replicates with the water conservation method (WCM), cropping systems, and randomized genotype treatments. *Local sorghum landrace Nongomsoba or local cowpea variety/improved sorghum variety Sariaso 14 or improved cowpea variety KVX 396-4-4.

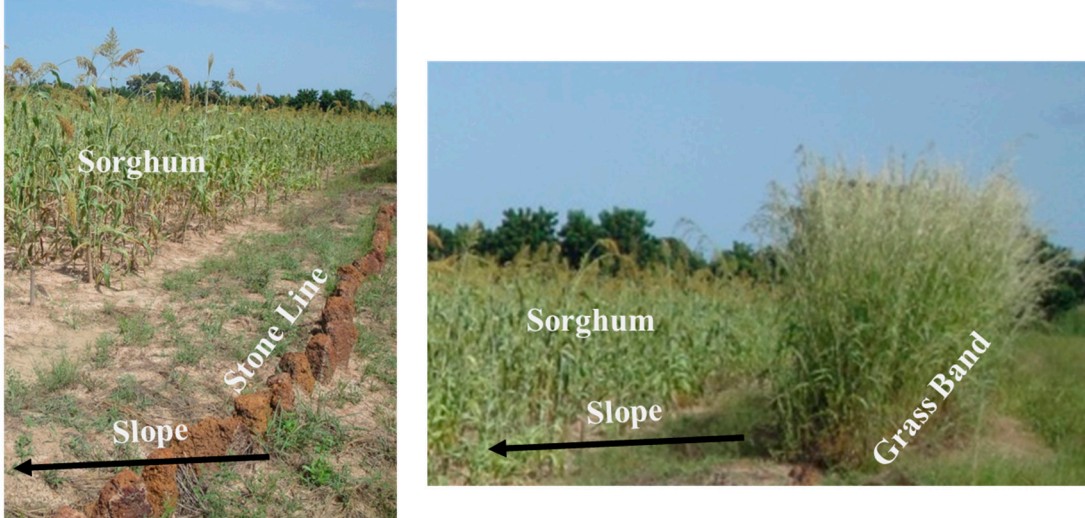

**Figure 3.** Layout of the stone line and grass band water conservation methods (WCM) at Saria Research Station, Burkina Faso.

*2.2. Data Collection and Statistical Analysis*

Starting 36 days after planting, soil water was measured every week in 2010, so the cumulative effect of three years of such practices could be evaluated. Soil samples were collected every 10 cm from 0 to 30 cm for three replications and were collected on a diagonal passing through the center of the plot. The wet weight was measured before drying in the oven. After drying the samples, the dry weight was measured and soil water (W%) was calculated as follows:

$$W\% = \{(\text{wet weight} - \text{dry weight})/\text{dry weight}\} \times 100.$$

Sorghum and cowpea grain and stover yields were measured annually at harvest from the center two rows of the subplots to avoid edge effects. Rainfall was measured and collected daily at a weather station. Economic analysis was done with 130 West African CFA franc (CFA) kg$^{-1}$ (i.e., 0.22 USD) for sorghum grain and 30 CFA kg$^{-1}$ (i.e., 0.05 USD) for sorghum stover. Cowpea grain price was 350 CFA kg$^{-1}$ (i.e., 0.58 USD) and 50 CFA kg$^{-1}$ (i.e., 0.08 USD) for cowpea stover. The USD to CFA

conversion rate used was 1:600. Data analysis was conducted using ANOVA in XLSTAT software XLSTAT Version 2016.02.28451. A Newman–Keuls test was used for mean separation at $\alpha = 0.05$.

## 3. Results

### 3.1. Soil Water Due to Water Conservation Method (WCM)

The WCMs (i.e., stone lines and grass bands) had no significant effect on soil water during early vegetative growth stages from August 10 to September 21 when rainfall distribution and quantity were adequate (Figures 1 and 4). Over 80% of the annual rainfall occurs during August and early September. In 2008 and 2009 there was adequate rainfall (i.e., over 800 mm with even distribution) throughout the growing seasons; however, after 29 September 2010, there was a significant reduction in rainfall at the end of the rainy season. During this period, a significant increase in soil water due to the WCM treatments was measured (Table 1; Figures 1 and 4). This led to significant changes in grain yield and revenue. Mean soil water for WCM were higher than that of the control. There was no significant difference between using stone lines or grass bands to increase soil water (Table 2; Figures 1 and 4). The effect of WCM treatments was only expressed in conditions of low rainfall or high water consumption by plants during grain fill and late reproductive stages. Additionally, soil water was significantly affected as a function of soil depth without interactions with WCM or varieties (Table 1). Shallower sampling depths showed a greater change in soil water, to a depth of 30 cm.

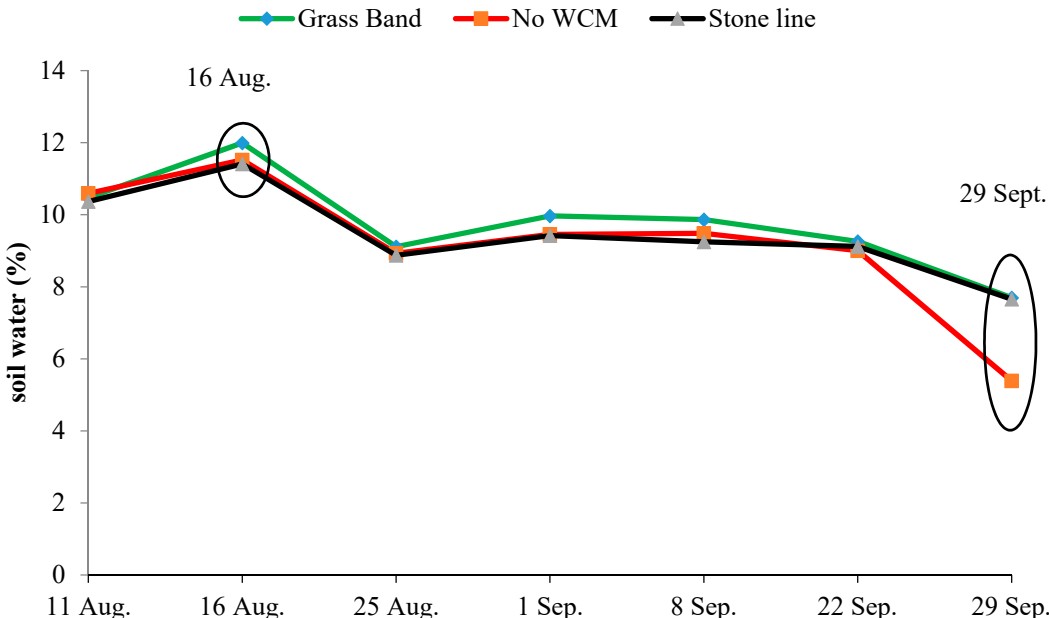

**Figure 4.** Soil water variation during the 2010 cropping season at Saria Research Station, Burkina Faso.

**Table 1.** Analyses of variance results (*probability*-values) for grain yield, soil water, and revenue.

| Factors | DF | Grain Yield | | | Soil Water | | Revenue | |
|---|---|---|---|---|---|---|---|---|
| | | 2008 | 2009 | 2010 | Vegetative Stages (16 August 2010) | Reproductive Stages (29 September 2010) | 2009 | 2010 |
| WCM | 2 | 0.824 | - | - | - | - | - | - |
| Varieties (First year) | 1 | 0.757 | - | - | - | - | - | - |
| WCM*Varieties | 2 | 0.780 | - | - | - | - | - | - |
| WCM | 2 | - | 0.790 | 0.2147 | 0.343 | <0.0001 | 0.420 | 0.225 |
| Varieties | 3 | - | <0.0001 | <0.0001 | 0.143 | 0.430 | <0.0001 | 0.0001 |
| WCM*Varieties | 6 | - | 0.206 | 0.940 | 0.607 | 0.101 | 0.723 | 0.877 |
| Soil Depth | 2 | - | - | - | <0.0001 | 0.001 | - | - |
| WCM*Soil Depth | 4 | - | - | - | 0.665 | 0.928 | - | - |
| Varieties*Soil Depth | 6 | - | - | - | 0.826 | 0.794 | - | - |

WCM, Water Conservation Method; DF, Degree of Freedom.

**Table 2.** Analyses of variance results for mean soil water, grains yield, and revenue.

| Factors | Grain Yield | | | Soil Water | | Revenue | |
|---|---|---|---|---|---|---|---|
| | (kg ha$^{-1}$) | | | 16 August (%) | 29 September (%) | (CFA ha$^{-1}$ \| USD ha$^{-1}$) | |
| Years | 2008 * | 2009 ** | 2010 * | 2010 | 2010 | 2009 | 2010 |
| Stone line | 1017 | 1079 | 1356 | 11.4 | 7.7 | - | 169,248 \| 282 |
| Grass band | 941 | 1033 | 1398 | 12.0 | 7.7 | - | 152,540 \| 254 |
| No WCM | 928 | 1105 | 897 | 11.5 | 5.4 | - | 133,672 \| 223 |
| *p*-value | 0.824 | 0.790 | 0.019 | 0.343 | <0.0001 | - | 0.225 |
| Standard Error | 109 | 74 | 125 | 0.299 | 0.213 | - | 14,276 \| 24 |
| Nongomsoba | 981 | 1707 | 590 | 11.6 | 7.2 | 330,610 \| 551 | 136,030 \| 227 |
| Sariaso 14 | 942 | 1062 | 309 | 12.0 | 6.7 | 243,981 \| 407 | 105,341 \| 176 |
| Local cowpea/Nongomsoba | - | 712 *** | 994 | 11.9 | 6.7 | 387,047 *** \| 645 | 215,147 \| 359 |
| KVX 396-4-4/Sariaso 14 | - | 808 *** | 518 | 11.0 | 7.0 | 468,535 *** \| 781 | 150,763 \| 251 |
| *p*-value | 0.7570 | <0.0001 | <0.0001 | 0.143 | 0.430 | <0.0001 | <0.0001 |
| Standard Error | 89 | 86 | 87 | 0.345 | 0.246 | 22,746 \| 38 | 16,485 \| 27 |

Bulk probabilities had significant different means. *: Sorghum only data; **: Sorghum and cowpea data; ***: Cowpea only data. No sorghum was grown in 2009 in the sorghum/cowpea rotation plots.

### 3.2. Crop Yield Due to WCM, Crop Rotation, and Variety

In 2008, there was no statistically significant difference in sorghum grain yield between the local and improved varieties corresponding with adequate rainfall. No cowpea was grown in 2008. In 2009, the first year of the rotation, there was a statistical difference between the cropping systems. Though continuous sorghum, regardless of variety, produced more grain yield than cowpea, cowpea production lead to significantly higher revenue per hectare in 2009 (Table 2). In 2010, when all plots were planted to sorghum, the sorghum–cowpea rotation had the highest significant grain yield and revenue (Table 2).

During the first year of the trial in 2008, there was no effect of the WCM treatments on sorghum grain yield, likely due to either adequate rainfall observed during the 2008 cropping season or due to a need for the grass bands and stone lines to be better established before their effects could be observed (Figure 5). Baseline sorghum yields across all treatments averaged 962 kg ha$^{-1}$. By 2010, the third year of experimentation, there was a significant difference observed between the effects of WCM for sorghum grain yield that corresponded with reduced rainfall and soil water (Figures 1 and 6). Sorghum grain yields with stone lines or grass bands were higher than that of the control by an average of 450 kg ha$^{-1}$ (Figure 6) which amounted to an increase of 58,500 CFA ha$^{-1}$ (i.e., 98 USD ha$^{-1}$).

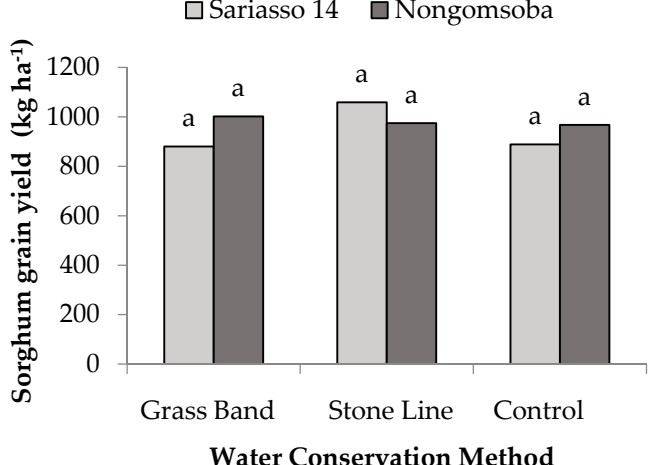

**Figure 5.** Sorghum grain yield of the two sorghum varieties as affected by water conservation methods at Saria Research Station, Burkina Faso, in 2008. Columns with the same letter are not significantly different at $p \le 0.05$.

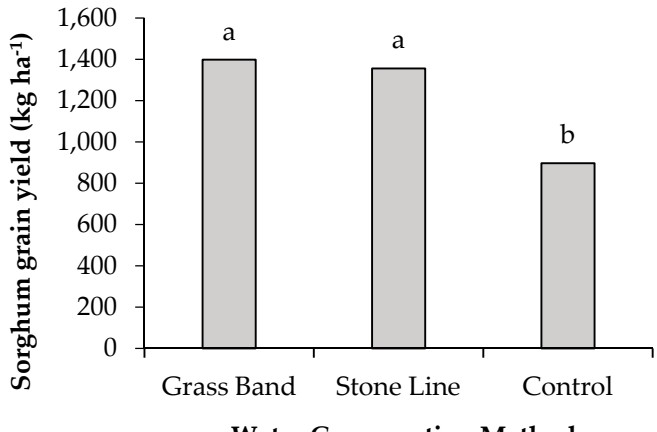

**Figure 6.** Sorghum grain yield as affected by water conservation methods at Saria Research Station, Burkina Faso, in 2010. Columns with the same letter are not significantly different at $p \le 0.05$.

Crop rotation had a significant positive effect on sorghum grain yield whereas improved varieties had a significant negative effect on sorghum grain yield in 2010 following one full crop rotation cycle. Local landraces using Nongomsoba sorghum in rotation with the local cowpea variety resulted in the highest sorghum yield as compared to the improved varieties of Sariaso 14 sorghum in rotation with KVX 396-4-4 cowpea (Figure 7). The effects of WCM were significantly more pronounced in higher-yielding situations where soil-water holding capacity was likely the primary yield-limiting factor. Continuous sorghum production using an improved variety (i.e., Sariaso 14) did not perform better than the local landrace even when in rotation. These differences were observed regardless of WCM. Grass bands consistently performed better than stone lines in each of the tested cropping systems though not significantly.

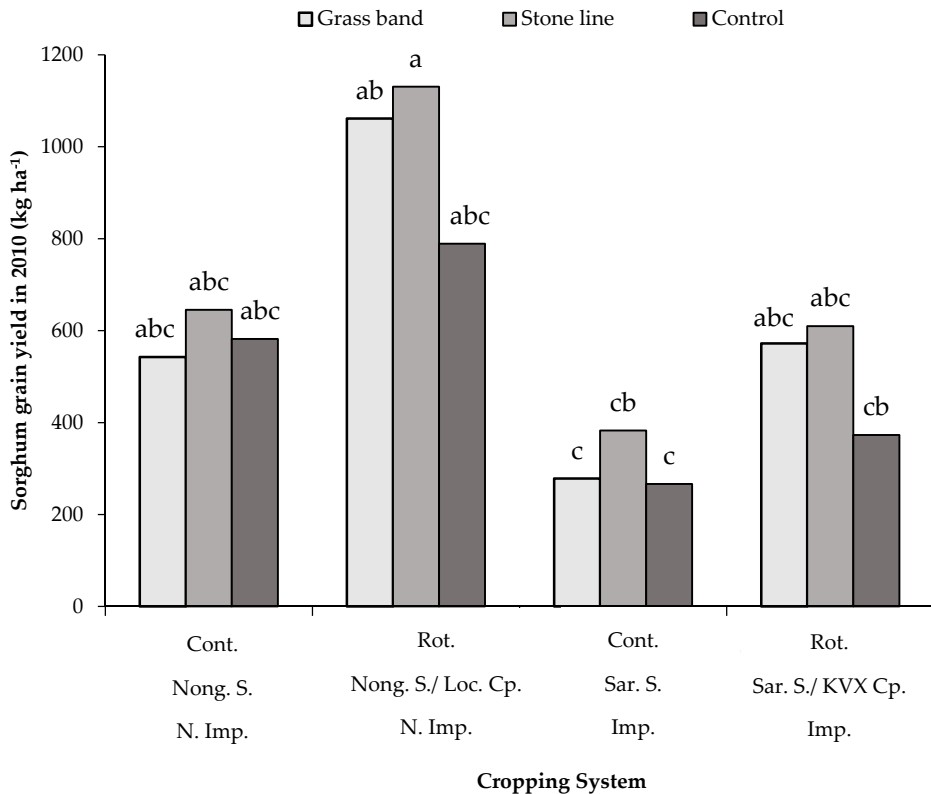

**Figure 7.** Sorghum grain yield of the two sorghum varieties in continuous cropping or rotation with cowpea according to WCM at Saria Research Station, Burkina Faso, in 2010. Abbreviations: Cont.: Continuous; Rot.: Rotation; Nong. S.: Nongomsoba Sorghum; Sar. S.: Sariaso 14 Sorghum; KVX Cp.: KVX 396-4-4 Cowpea; Loc. Cp.: Local Cowpea; N. Imp.: Nonimproved; and Imp.: Improved. Columns with the same letter are not significantly different at $p < 0.05$.

### 3.3. Revenue Due to WCM, Crop Rotation, and Variety

Even though there was no significant difference in cowpea grain yield between the local and improved cowpea varieties, grain and stover revenue for the improved cowpea variety KVX 396-4-4 returned the greatest revenue due to additional stover (Table 2; Figure 8). The improved cowpea variety had significantly more revenue per hectare than the local cowpea variety and the sorghum varieties at 468,535 CFA ha$^{-1}$ (i.e., 781 USD ha$^{-1}$). The improved sorghum variety, Sariaso 14, had the lowest grain and stover revenue at 243,981 CFA ha$^{-1}$ (i.e., 407 USD ha$^{-1}$) (Table 2; Figure 8). In 2009, the highest incomes were obtained on plots with cowpea variety KVX 396-4-4 (i.e., 468,535 CFA ha$^{-1}$ or 781 USD ha$^{-1}$) and are significantly higher than that of the local cowpea variety by 21%. For sorghum, the local sorghum variety Nongomsoba generated an income of 330,610 CFA ha$^{-1}$ (i.e., 551 USD ha$^{-1}$), which was 36% more than the improved sorghum variety Sariaso 14 (Figure 8). For the local Nomgomsoba sorghum in rotation with local cowpea, the income gap with the continuous cropping plot was 43%–58% (Figure 9). The improved sorghum variety (i.e., Sariaso 14) generated the lowest income due to low yield. In 2010, the rotation of the nonimproved cowpea and sorghum varieties had significantly higher revenue than continuous sorghum, both for improved and nonimproved varieties, and as compared to improved sorghum and cowpea varieties in rotation (Table 2; Figure 9). In general, cowpea was more profitable than sorghum per hectare, regardless of the crop genetics.

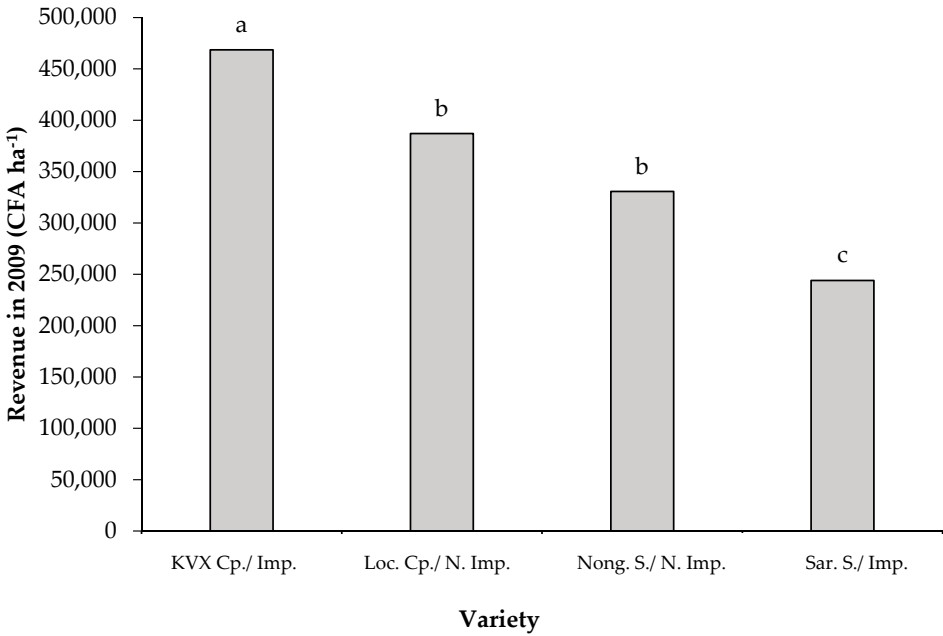

**Figure 8.** Mean revenue for cowpea or sorghum grain and stover in 2009. All plots were cropped in sorghum in 2008 and 2010. In 2009, for the crop rotation plots, a local cowpea variety was grown after Nongomsoba (local sorghum) while the improved cowpea variety KVX 396-4-4 was used after Sariaso 14 (improved sorghum). Abbreviations: KVX Cp.: KVX 396-4-4 Cowpea; Loc. Cp.: Local Cowpea; Nong. S.: Nongomsoba Sorghum; Sar. S.: Sariaso 14 Sorghum; Imp.: Improved; and N. Imp.: Nonimproved. Columns with the same letter are not significantly different at $p < 0.05$.

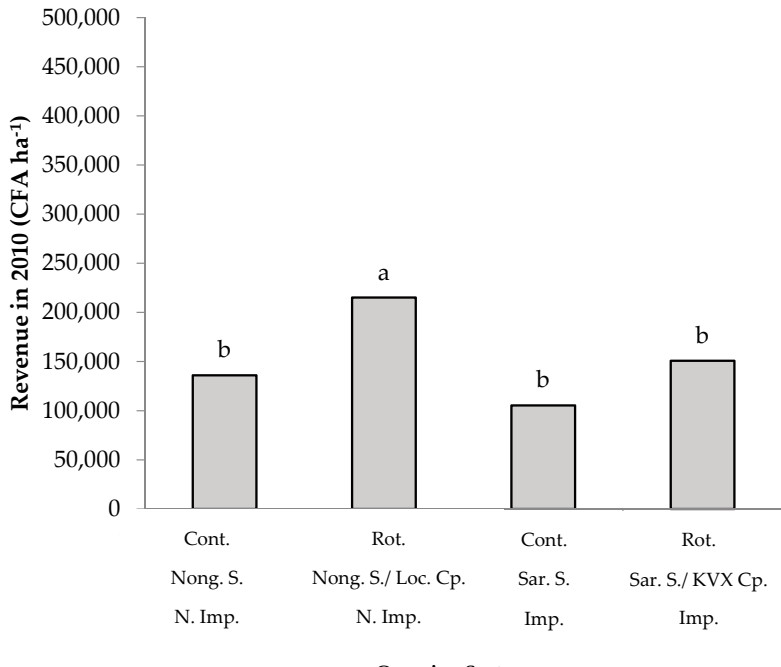

**Figure 9.** Mean grain and stover revenue for continuous sorghum or sorghum in rotation with cowpea. Sariaso 14 sorghum and KVX 396-4-4 cowpea are improved varieties, whereas Nongomsoba sorghum and the local cowpea have not been improved through breeding. Abbreviations: Cont.: Continuous; Rot.: Rotation; Nong. S.: Nongomsoba Sorghum; Sar. S.: Sariaso 14 Sorghum; KVX Cp.: KVX 396-4-4 Cowpea; Loc. Cp.: Local Cowpea; N. Imp.: Nonimproved; and Imp.: Improved. Columns with the same letter are not significantly different at $p \leq 0.05$.



## 4. Discussion

WCM has little effect on soil water, grain yield, stover yield, and revenue in years with both adequate quantity and distribution of rainfall during the growing season, as occurred in 2008 and 2009. However, in years with erratic rainfall as documented in 2010, stone lines and grass bands increase soil water and resulted in increased grain yield, stover yield, and revenue. The lack of soil water at the end of the growing season can lead to reduced yields which is in agreement with [23,24]. However, this study showed that WCM can increase soil water and reduce grain yield loss during periods with reduced rainfall. Surface runoff is a major cause of water loss and soil degradation in semiarid rain-fed agricultural systems [15]. Stone lines and grass bands, which are permeable barriers, induce more surface water storage and infiltration [25] that can help improve sorghum yield when the nutrient needs for production have also been met. Thus, during periods with reduced rainfall, WCM is an effective resilience tool and "climate-smart technology" for preserving productivity and revenue. Reduced rainfall periods are not rare in the region as the north Sudanian climate is characterized by monomodal rainfall with extreme rain and drought periods. With future increase in climate variability due to climate change, which is expected and currently occurring [9], stone lines and grass bands will likely have growing importance as a resilience strategy to buffer crop productivity and revenue during periods of too much or too little rainfall. Additionally, productivity and revenue of the local varieties was further enhanced when paired with WCM and outperformed both continuous cropping of sorghum, regardless of variety, and rotation of improved sorghum and cowpea (Figure 9).

The local sorghum variety (i.e., Nongomsoba) significantly outperformed the improved and released Sariaso 14 variety in terms of both productivity and revenue. The trial location was a relatively low input and productivity environment, similar to most smallholder farmer fields in Burkina Faso. Also, the yield difference may have been driven by the root system of the local landrace Nongomsoba, because it is more developed than that of the improved Sariaso 14 variety [22]. However, under higher soil fertility scenarios, the improved variety may outperform the local variety. In the case of cowpea production, there was no difference between the local and improved variety (i.e., KVX 396-4-4). These findings are consistent with the fact that there is relatively low adoption of improved sorghum and cowpea varieties in Burkina Faso and most of the Sudano–Sahelian region, especially when considering the reduced access and increased costs associated with improved varieties. This finding is corroborated by other varietal adoption studies across sub-Saharan Africa [26–28].

There were additional productivity and revenue benefits associated with crop rotation, which was observed after one cropping cycle (Figure 9). Starting in 2004, Burkina Faso developed a rural development strategy that took into account crop diversification, including cowpea [29]. This study showed that crop diversification has an additive effect with WCM as a resilience strategy for nutritional food security and income security and is consistent with others [30]. In both cowpea production years and sorghum following cowpea years, grain and stover revenue was significantly higher in the crop rotation system (Figures 8 and 9). Though the grain yield of cowpea per hectare was significantly less than that of sorghum, the increased value of cowpea grain, which is 2.7 times higher than that of sorghum per hectare (i.e., 468,535 CFA ha$^{-1}$ or 781 USD ha$^{-1}$), still led to higher revenue per hectare (Figure 8). Increased revenues for the sorghum following cowpea years were likely due to enhanced soil fertility from the incorporation of cowpea, an N-fixing legume.

Legumes are vital for sustainable production because they directly supply high C: N organic material through biological nitrogen fixation, and can also be an indirect source of manure-based nitrogen inputs to maintain soil productivity [31]. Cowpea in rotation with sorghum also improves crop diversification to reduce pest pressure and improve food security and nutrition as cowpea is a good source of protein and other essential vitamins and minerals. As a short cycle crop (i.e., 60 day maturation period), the integration of cowpea into cropping systems also contributes to improved food and nutrition resilience by providing food during the "hunger period" when food stocks are low even as crops are growing in the field but are not ready for harvest. Pulse crops have an important role in sustainable agriculture, agroecology, family nutrition, and income in sub-Saharan Africa [32].

Cowpea provides needed dietary diversity, and are key contributors to human nutrition as a crucial source of protein, amino acid diversity, and B-group vitamins, iron, zinc, magnesium, and calcium [33]. Cowpea fodder is also highly nutritious for livestock production, which provides income diversity and another route for improved human nutrition from animal source proteins [34]. The role of grain legumes in African agricultural systems is multifaceted and often provides an important source of income as they can be sold for high prices at local or international markets [32].

**5. Conclusions and Recommendations**

Resilience of smallholder farmers in their ability to bounce-back and overcome shocks is critical to ensuring a pathway out of hunger and poverty, reducing the number of displaced people, and has been associated with reducing conflict and the rise of insurgency groups [35]. In this study, we show that stone lines or grass bands and crop rotation are effective resilience strategies individually and in combination. Plots with stone lines or grass bands stored more water than the control. This study showed that using stone lines or grass bands is effective in enhancing soil water content on degraded land. During years in which rainfall is well-distributed over time, there is no difference between fields with WCM and fields without. However, when there are periods of prolonged drought, WCM is an effective method for increasing soil water, yield, revenue, and resilience. The results also suggest that the combination of WCM and crop rotation has an additive effect on improving cropping systems productivity and revenue. Growing cowpea in rotation with sorghum diversifies production and gives more options to farmers to increase their incomes. This study also sheds light on limited productivity gains due to improved crop varieties. Low input and degraded soils, even under scenarios utilizing improved agronomy, often cannot support the genetic potential of improved crop varieties. Under low input/degraded conditions, improved crop varieties likely are not a suitable resilience strategy alone. This is consistent with the overall low adoption rates of improved crop varieties across the Sahelian agriculture. We conclude that, as extreme rainfall events are rising and are projected to continue to increase in the Sudano–Sahelian region, stone lines or grass bands in combination with sorghum and cowpea rotation will be effective practices for increasing resilience of smallholder farmers to maintain crop productivity and revenue. This study also showed that using stone lines or grass bands is an effective strategy to enhance soil water on degraded land.

This study did not address the labor and economic costs associated with installing stone lines and grass bands that should be evaluated in future work. In addition, research on the durability of stone lines on different soil types and the effectiveness of stone lines and grass bands on soil quality, water quality, and erosion should be evaluated to show the full benefit of these technologies.

**Author Contributions:** Conceptualization, H.T., A.B., D.Y., and V.P.; methodology, H.T., A.B., D.Y., and V.P.; formal analysis, H.T., A.B., and D.Y.; investigation, all authors; funding acquisition, H.T., A.B., D.Y., and V.P.; writing—draft preparation, all authors; writing—review and editing, all authors; supervision and project management, H.T., A.B., D.Y., and V.P. All authors have read and agreed to the published version of the manuscript.

**Funding:** This manuscript is made possible by the support of the American People provided to the International Sorghum and Millet Collaborative Research Support Program (INTSORMIL CRSP) and the Feed the Future Innovation Lab for Sustainable Intensification (SIIL, Cooperative Agreement No. AID-OAA-L-14-00006) through the United States Agency for International Development (USAID). The contents are the sole responsibility of the authors and do not necessarily reflect the views of USAID or the United States Government or institutions of the authors.

**Acknowledgments:** The authors are grateful to Institut de l'Environnement et de Recherches Agricoles (INERA) of Burkina Faso for facilitating the implementation of the trial at the Saria Research Station. Contribution no. 20-240-J from the Kansas Agricultural Experiment Station.

**Conflicts of Interest:** The authors declare no conflict of interest. The funders had no role in the design of the study; in the collection, analyses, or interpretation of data; in the writing of the manuscript, or in the decision to publish the results.

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
