# Peer review of "Water Conservation Methods and Cropping Systems for Increased Productivity and Economic Resilience in Burkina Faso"

_water, doi:10.3390/w12040976_

Round 1

Reviewer 1 Report

This is an interesting field study on the combined effect of water conservation measurements, cropping systems and grain variaties on the grain productivity/economin revenue of farming systems in Burkina Faso.

In generall, the introduction is well written. However, objectives need improvement.

The methodology should be improved, e.g. a schematic figure of the experimental setup, information on rainfall data aquisition.

Presentation of results need improvement. Figures and Tables need to be improved. Rainfall data and soil water content data from 2008 and 2009 is not presented. 

Discussion and Conclusions are well written and sound. However, no mention of future work was made.

Some comments:

. L19-20: "During years where rainfall is well distributed over time, there is no difference between fields with water conservation methods and fields without." Differences may be minor to almost nothing, however some differences may still exist. So instead of "there is no difference" maybe use e.g. "differences are minimal". 

. L76-78: Objectives should be improved. They need to be more detailed.

.L91: Add units of bulk density.

.The explanation of the experimental design is clear. However, a figure with a scheme (plots, sub-plots and so on) would really improve the reading.

.L115: Why was the soil water only measured in 2010.

.L120: "Sorghum and cowpea grain and stover yields were measured..." When? For the three seasons?

.L124-126: This information is repeated.

.L135: "In 2008 and 2009 there was adequate rainfall throughout the growing season." These data is not presented. How was rainfall data collected?

.The data/results are confusing. In Table 2, soil water is presented only for 2010, grain yeld is presented for 2008, 2009 and 2010, and then revenue is presented only for 2009 and 2010 (and not for the different WCM).

.The black circle (16 Aug) in fig. 2 belongs to Fig. 1.

.What variety of sorghum and cropping system is presented in Fig. 4? Because, apparently, the same bars are shown in Fig. 5, however they do not match. Also, Fig. 4 appears in the text after Figs. 5 and 6.

Reviewer 2 Report

Comments on “Water Conservation Methods and Cropping Systems for Increased Productivity and Economic Resilience in Burkino Faso

Submission to Water, March 2020

General

This seems to me a competent account of a useful study, though diminished in impact by the rather old data (from 2008-2010) and by excessive wordiness, including some repetition.

And Figure 3 is missing (or mis-numbered) in the review file

Most of the specific comments below are of minor significance, but attention to them will improve the readability of the paper

Specific (by page and line number in review file)

1, 8: Kansas

1, 17: terms such as “this study” do not mean much when the reader has not yet been introduced to the study (or experiment?). And sometimes, “this paper” is meant.

1, 19: “when”, not “where”

1, 24: these absolute amounts may not mean much: what about percentages?

1-2, 40-48: SSA, “West Africa” and “the Sudan to Sahel regions” (plural) are all mentioned, which is confusing: do they overlap, and in which area does Burkino Faso lie?

2, 49: insert “by” before “the ability”

2, 53: better as “… 2017), giving increased urgency to …”

2, 53: the phrase “soil water holding capacity” is unclear without hyphens

2, 61: enter hyphens in “high-yielding” and “stress-tolerant” here and elsewhere in the paper

2, 66: no need for both “Additionally” and “also”

2, 68: “these” can replace several repeated words

2, 73-76: rather a long sentence, with “in theory” further obscuring the grammar

2, 81: as at (1, 17): the reader does not yet know about “the” trial

2, 89: insert “the” (or “a”) before “USDA”

2, 91: what are the units of “1.7”? and “within” rather than “from” seems more suitable

3, 97: insert hyphen in “sub-factors”

3, 109: better “from 115 to 120”, or “over 115-120”, and similarly elsewhere

3, 111: “only data of that year [2010] were considered”. But many later references (including some in tables) refer to 2008 and/or 2009. Perhaps add explanatory words after “considered”

3, 118: why “rate”, which is usually a speed ratio or similar, not a %?

3, 123-124: presumably USD:FCFA is 1:600, so reverse one pair

3, 124-126: these lines repeat exactly (2-3, 94-97): delete one

3, 128: “A Newman-Keuls test was used ..”

3, 130: in subsection heading, use “water conservation method” in full (with “WCM” to follow): the acronym is otherwise buried (twice) in the previous section

3, 133: “were” not “was”

3, 134: “Over 80% of the annual rainfall occurs during August and early September”

3, 136-137: what is the cause of higher “soil water” (“rates”, content?) in 2010? - the sentence appears to promise rainfall different in 2010 than in 2008 and 2009, but instead refers to (“due to”) the WCM treatments. Was rainfall significantly different (lower, presumably) in 2010?

3, 137: “led” (the past tense) not “lead”

3, 139: why is Table 2 referred to before Table 1?

4, 145-146: column width needs adjustment to avoid distorted headings

4, 144 and 146: why is the order of “grain(s?) yield” and “soil water” (rate?) different in the two table titles?

4, Table 2: the use of 6 digits in FCFA results clutters up tables, and is pointless. Use ‘000 FCFA throughout?

4, 147: I do not understand the “NB” note

5, 151: does the ellipse below “16 Aug.” mean anything? Would a downward arrow suffice to indicate one (of several? – see (3, 132)) “sampling date”?

5, 156: by “statistical difference”, I think you mean “statistically significant difference”; maybe former phrase OK later (e.g. at 158), but spell out in full here at first use

6: these Figures 5 and 6 are located out of order – they come before Figure 4 on page 7. And, where is Figure 3, referred to at (7, 176)?

6, Figure 5: explain “abc” etc. in the diagram. Also, column labels are rather cryptic

6, 163: “continuous”

6, Figure 6: as mentioned above, the multiple “000” are distracting and unnecessary: switch to ‘000 FCFA

6, 167-170: this is a very long title (or note) to the table; much should be (and perhaps is) in the main text

7, 174: insert comma after “yield”

7, 176: where is Figure 3, and why is it referred to here, after references to Figures 4-6?

7, 177: “… difference observed between the effects of WCM …”, or similar

7, 178: to what does “which” refer? – not “yield” or “WCM”, I suppose

7, 179: better “… with stone lines or with grass bands …”, since both not used together

7, 183: “affected”, not “effected”

7, 188: “effects” plural, I think; and insert hyphen after “higher” (as also elsewhere, perhaps)

7, 199: insert “at” after “revenue”

8, Figure 7: column labels (and horizontal axis) are distorted in the review file; see also above for ‘000 and “a”, “b”

8, 212: “continuous”; and Figure title is again too long

8, 217-221: this paragraph is oddly ordered, with the first sentence reporting (I presume) experimental results, the second repeating (? – see Section 1) region-wide description, and the third returning to “this study”. The fifth sentence (“rainy season”) gives (repeats?) experimental context. Decide what is to be “discussed” (and not just reported as in Sections 3 and 5), and discuss in logical manner.

8, 229: again, what is the logical order here? – the paragraph starts similarly to the one above, and moreover with a “result” (“WCM improved … yield”) that seems to underlie a “conclusion” previously stated, before referring to previous (and rather old) papers (which may be a focus of “discussion”).

9, 236: “productivity”

9, 237: better “outperformed those [pd’y and rev] obtained from continuous …”

9, 240: this paragraph starts with a very general point (already made in Section 1?): what is to be “discussed” here? – presumably local vs. improved varieties. If so, make clearer

9, 243: does “This” refer to the local variety (and not the improved one, also mentioned?

9, 244: again “this difference” is not clear, partly since no “difference” in previous sentence

9, 248: “varieties” plural, and “findings” plural (if more than one?)

9, 254: delee comma after “Faso”

9, 257: what is the subject of “is supported by”? – not “this study”, presumably

9, 261: not quite clear if the numbers are differences or levels (if the former, what %?); and “led” not “lead”

9, 263: “the N-fixing legume” (i.e. cowpea, presumably)

9, 265: “biological”, I think – does it refer to “nitrogen” or to fixation”?

9, 270: this is getting repetitious, with “resilience” following “security”

9, 281: little in this section (no. 5?) is new, so reduce its lengthy, and/or earlier statements of the same things

9, 283: “ensuring”; also verbs in this line are many: what is the subject of each?

9, 286: but you did not “combine” stone lines and grass bands; make this clearer

10, 287: “… is effective in enhancing soil water [content] …”

10, 290: “suggest”

10, 294-295: sentence structure needs improvement/correction: what is the subject of “often cannot”?

10, 296: why “similar” (to what?); delete?

10, 299: insert comma after “that”; and, events do not “rise”, they “occur”; re-phrase

Reviewer 3 Report

Comments are in the attached file.

Round 2

Reviewer 1 Report

The reviewed manuscript is much improved and the authors answered satisfatorilly to all comments and queries to the original manuscript.

Therefore, I recommend publication of the manuscript in its present form.

I simply suggest to move the future work statements (L327-331) to the end of the "Conclusions and recommendations" section, in a separated paragraph, after all conclusions were made.

Reviewer 3 Report

The comments are in the file attached.
